Distribution and characteristic of nitrite-dependent anaerobic methane oxidation bacteria by comparative analysis of wastewater treatment plants and agriculture fields in northern China

Hu Zhen huzhen885@sdu.edu.cn
Ma Ru
School of Environmental Science and Engineering, Shandong University , Jinan , China
Liu Yu
Electronic publication date: 2016 Dec 14
Publication date: 2016
Volume: 4
Electronic Location ID: e2766
Received 2016 Jun 20; Accepted 2016 Nov 7
Copyright: ©2016 Hu and Ma
Copyright year: 2016
Copyright holder: Hu and Ma
License: This is an open access article distributed under the terms of the Creative Commons Attribution License, which permits unrestricted use, distribution, reproduction and adaptation in any medium and for any purpose provided that it is properly attributed. For attribution, the original author(s), title, publication source (PeerJ) and either DOI or URL of the article must be cited.
License URL: https://creativecommons.org/licenses/by/4.0/

Keywords: Nitrite-dependent anaerobic methane oxidation, Greenhouse gas, Comparative analysis, Distribution, Northern China

Funding: National Natural Science Foundation of China 21307076 Fundamental Research Funds of Shandong University 2014TB003 Seventh Framework Programme FP7-PEOPLE-2011-IRSES Funding was provided by the National Natural Science Foundation of China (No. 21307076), Fundamental Research Funds of Shandong University (No. 2014TB003), and the Seventh Framework Programme FP7-PEOPLE-2011-IRSES. The funders had no role in study design, data collection and analysis, decision to publish, or preparation of the manuscript.

==============================
Nitrite-dependent anaerobic methane oxidation (n-damo) is a recently discovered biological process which has been arousing global attention because of its potential in minimizing greenhouse gases emissions. In this study, molecular biological techniques and potential n-damo activity batch experiments were conducted to investigate the presence and diversity of M. oxyfera bacteria in paddy field, corn field, and wastewater treatment plant (WWTP) sites in northern China, as well as lab-scale n-damo enrichment culture. N-damo enrichment culture showed the highest abundance of M. oxyfera bacteria, and positive correlation was observed between potential n-damo rate and abundance of M. oxyfera bacteria. Both paddy field and corn field sites were believed to be better inoculum than WWTP for the enrichment of M. oxyfera bacteria due to their higher abundance and the diversity of M. oxyfera bacteria. Comparative analysis revealed that long biomass retention time, low NH4+ and high NO2− content were suitable for the growth of M. oxyfera bacteria.

Introduction

Methane (CH4) and nitrous oxide (N2O) are important greenhouse gases, accounting for about 20% and 7% of global warming respectively (Griggs & Noguer, 2002; Knittel & Boetius, 2009). Cai (2012) reported that anthropogenic activities, rather than natural sources, are the major sources of CH4 and N2O emissions. It is widely accepted that wastewater treatment plants (WWTPs) and agricultural fields are two of the most important anthropogenic GHGs sources (Foley et al., 2011; Liu et al., 2014a). In WWTPs, enormous amount of CH4 andN2O would be produced during the biological transformation of carbohydrates and nitrogenous compounds, respectively. Our previous on-site investigation showed that about 4.48–9.68 × 109 g of CH4 and 0.93–1.28 × 109 g of N2O would be emitted from WWTPs of China per year (Wang et al., 2011a; Wang et al., 2011b). Compared with WWTPs, agricultural field was believed to be more important GHGs sources, mainly because the widely usage of chemical fertilizers to improve the productivity (IPCC, 2001). It is reported that agriculture field would contribute to 60% of N2O and 50% of CH4 emissions on a global scale (Montzka, Dlugokencky & Butler, 2011; Syakila & Kroeze, 2011).

Anaerobic methane oxidation (AMO) is a recently discovered sink of methane on Earth, with a consumption rate of approximately 70–300 Tg CH4 year−1 globally (Cui et al., 2015; Hu et al., 2011). Besides AMO coupled to reduction of sulfate (Timmers et al., 2015; Bian et al., 2001), humic compound (Smemo & Yavitt, 2007), iron (Beal, House & Orphan, 2009; Segarra et al., 2013) and manganese (Egger et al., 2015), the coupling of AOM to nitrite reduction process, named as nitrite-dependent anaerobic methane oxidation (n-damo), has also been demonstrated (Raghoebarsing et al., 2006) N-damo process is performed by “Candidatus Methylomirabilis oxyfera” (M. oxyfera) bacteria, which is affiliated with the NC10 phylum (Ettwig et al., 2010). N-damo process established a unique relationship between carbon cycle and nitrogen cycle (Raghoebarsing et al., 2006) and it was believed to be a promising method to minimize greenhouse gases emissions through converting CH4 and N2O to CO2 and N2 respectively (Raghoebarsing et al., 2006; Shen et al., 2015).

Presently, many studies have focused on the distribution of M. oxyfera bacteria in natural environments, e.g., freshwater lakes (Liu et al., 2014b) paddy soil (Wang et al., 2012) marine sediments (Chen, Jiang & Gu, 2014), wetlands (Hu et al., 2014) and so on. However, to date, information about distribution of M. oxyfera bacteria in northern China is still lacking. In addition, various inoculums have been reported to be able to enrich M. oxyfera bacteria successfully, including freshwater sediment (Raghoebarsing et al., 2006) sewage treatment sludge (Luesken et al., 2011a) ditch sediments (Ettwig et al., 2009) and paddy soil (Shen et al., 2014; Wang et al., 2012). He et al. (2014) found that inoculum sources had significant effect on enrichment of M. oxyfera bacteria, and claimed that paddy soil was the optimal inoculum. However, intensive study on inoculum sources from the perspective of microorganism is absence.

In this study, the presence and diversity of M. oxyfera bacteria in four different sites of northern China (paddy field, corn field, wastewater treatment plant (WWTP) and n-damo enrichment culture) were investigated through molecular biological techniques and potential n-damo activity batch experiments. Comparative analysis of environmental features and M. oxyfera bacteria activity was conducted to reveal the characteristics of M. oxyfera bacteria, as well as its optimal growth conditions.

Materials and Methods

Site description and sample collection

Non-flooded paddy fields with rice reaping once per year (PF) and corn fields with maize-wheat rotation for over 50 years (CF), both of which are typical agricultural type of northern China, were selected as agricultural field sample sites. PF cores and CF cores were collected from three locations (5 m distance) at the 50–60 cm depth in each sampling site, according to the previously described methods (Hu et al., 2014). Sludge from the anaerobic tank of a local WWTP (Everbright Water, Jinan China) (WS), and a lab-scale Upflow Anaerobic Sludge Bed reactor (UASB) aiming at enrichment of M. oxyfera bacteria (EC), were selected as WWTP samples. The sample collection was conducted in October, 2015, and the environmental characteristics of each sample site are listed in Table 1.

Table 1 Environmental characteristics of the sample sites.

Sample sites	Geographic coordinates	Temperature (°)	pH	Ammonium (mg N/kg dry sed)	Nitrite (mg N/kg dry sed)	Nitrate (mg N/kg dry sed)	Salinity (‰)	
PF	N36°41′, E116°54′	17	7.3	10.34	0.75	26.97	1.8	
CF	N37°44′, E115°40′	15	7.0	2.627	0.37	46.44	1.1	
EC	N36°40′, E117°03′	32	7.0	0.125	14117.65	941.18	1.2	
WS	N36°42′, E117°02′	22	7.6	815.88	127.19	735.29	2.1	

All collected samples were placed in hermetic containers and immediately transported to the laboratory within 4 h. Subsequently, the collected samples were equally divided into three parts. The first part was placed in the incubator to measure the potential n-damo activity, the second parts was stored in refrigerator at 4 °C for analysis of physicochemical parameters, and the last part was stored in refrigerator at −20 °C for further microbiological  analysis.

Physicochemical parameters analysis

Soil samples were extracted with 1M KCl and the concentrations of ammonium, nitrite and nitrate were measured as described by Ryan, Estefan & Rashid (2007). Soil pH was measured at soil/water ratio of 1:2.5 using a pH analyzer (HQ30d 53LEDTM, HACH, USA) (Wang et al., 2012). Temperature and salinity of soil was measured in situ using HI98331 soil electrical conductivity meter (HI98331, HANNA, Shanghai).

Concentrations of ammonium, nitrite and nitrate in water samples were analyzed according to the standard method (APHA, 2005). Water temperature, pH and salinity were measured in situ using pH and salinity analyzer (DDBJ-350; Leici, Shanghai). The CH4 concentration in the gas phase was analyzed using a gas chromatograph equipped with a flame ionization detector (FID–GC) (7890B; Agilent Technologies, Santa Clara, CA, USA).

Potential n-damo activity batch experiment

All the samples were washed three times with anaerobic water to remove the residual NOx− (NO2− and NO3−) an dorganic compounds, and were then transferred to 1L Ar-flushed glass bottles. The slurries were pre-incubated under anoxic conditions at 32 ± 1 °C for at least 48 h, in order to let the microbes adapt to the new environment. The bottle was flushed with Ar gas again before the measurement of potential n-damo activity. Two treatment groups were conducted subsequently: (a) CH4 (blank group), (b) CH4 + NO2− (experimental group). The initial CH4 concentrations in both blank and experimental groups were 1.02 ± 0.06 mmol L−1 and the initial concentrations of NO2− in the experimental groups were 0.35 ± 0.01 mmol NO2− L−1. The variation of CH4 and NO2− concentrations were determined at intervals of 6 h. The potential methane oxidation rates and the ratio of CH4/NO2− were evaluated by linear regression of CH4 and NO2 decrease in the experimental groups.

Fluorescence in situ hybridization (FISH)

Approximately 0.3 g of collected samples were washed in phosphate-buffered saline (PBS; 10 mM Na2HPO4/NaH2PO4 pH 7.5 and 130 mM NaCl) and fixed with 4% (w/v) paraformaldehyde in PBS for 3h under 4∘. After incubation, the sediment (fixed biomass) was washed with PBS and then stored in mixture (1 ml) of ethanol and PBS (1×) at −20 °C until analysis.

Bacterial probe S-*-DBACT-1027-a-A-18 (5′-TCTCCACGCTCCCTTGCG-3′) (Cy3, red), specific for bacteria affiliated with the NC10 phylum, were used in this study (Raghoebarsing et al., 2006) and a mixture of EUB I-III (FITC, green) was used for the detection of total bacteria (Daims et al., 1999). Fixed biomass (10 µl) was spotted on microscopic slides circles and then dehydrated subsequently with 50%, 80%, and 98% of ethanol for 3 min each. The probes were hybridized for 2 h at 46 °C in hybridization buffer (5M NaCl, 1M Tris/HCl pH 8.0, 10% sodium dodecyl sulfate) and 40% formamide. Hybridized samples were washed with hybridization leachate at 48 °C and then added with the fluorescence decay resistance agent. A fluorescence microscope (BX53; Olympus, Tokyo, Japan) was used to observe the prepared slides and the picture was disposed with Image-Pro Plus 6.0.

DNA extraction and PCR amplification

Total DNA was extracted using a Power Soil DNA Isolation kit (Mo Bio Laboratories, Carlsbad, CA, USA) according to the manufacturer’s protocols. The DNA concentration was measured at 260 nm with a NanoDrop spectrophotometer (NanoDrop 1000; Nano-Drop Technologies, USA).

To understand the biodiversity of M. oxyfera bacteria, 16S rRNA gene of M. oxyfera bacteria was amplified using nested PCR protocols, as previously described (Hu et al., 2014). In nested PCR, products of the first round PCR were then used as the DNA templates in the following round PCR. For 16S rRNA gene amplification, the specific forward primer 202F (Ettwig et al., 2009) and general bacterial reverse primer 1545R (Juretschko et al., 1998) were used for the first round PCR, and NC10 specific primers qP1F and qP2R (Ettwig et al., 2009) were performed for the second round PCR. The detailed information of nested PCR is shown in Table S1.

Quantitative real-time PCR (qPCR)

The quantitative PCR of M. oxyfera bacteria 16S rRNA gene were performed on LightCycler480 with Sequence Detection Software v1.4 (Applied Biosystems, Life Technologies Corporation, USA). The abundance of 16S rRNA gene was determined using the primers qp1R-qp1F (Ettwig et al., 2009) with 10 µL of Power SYBR Green PCR Master Mix, 1 µL of template DNA (5–20 ng µL−1), 0.4 µL of each primer and 8.2 µL of ddH2O. Detailed information is exhibited in Table S1. Negative-control reactions in which the DNA template was replaced by nuclease-free water were also performed. The whole process was performed under sterile conditions on ice and away from light. Triplicate qPCR analyses were performed for each sample. The standard curve was constructed from purified plasmid DNA with the concentrations ranging from 1.0 × 101 to 1.0 × 107 copies µL−1, and it showed correlation between the DNA template concentration and the crossing point with coefficients of determination (R2 > 0.97). The qPCR amplification efficiency of the standard curve and reactions were both greater than 85%.

High-throughput pyrosequencing and data analysis

After amplification, the purified nested PCR products of 16S rRNA gene were used for pyrosequencing on the Roche 454 GS-FLX Titanium sequencer (Roche 454 Life Sciences, Branford, CT, USA) at Personalbio (Shanghai Personal Biotechnology, Co., Ltd., Shanghai, China).

After pyrosequencing, all the raw reads were analyzed using QIIME standard pipeline (Shu et al., 2015) to trim off the low quality reads, adaptors, barcodes and primers. Then, sequences containing ambiguous base calls (Ns) and sequences shorter than 150 bp were also removed. The remaining sequences were clustered into operational taxonomic unites (OTUs) by UCLUST, with identity of 97% (Edgar et al., 2011). The sequences with highest relative abundance in each OUT were annotated with NCBI taxonomy using BLASTN and the Green genes database. Chao1 richness estimator, ACE estimator, Simpson diversity and Good’s coverage were calculated in Mothur analysis (http://www.mothur.org). Beta diversity statistics, including cluster analysis, weighted UniFrac distance metrics, and Principal coordinate analysis (PCoA), were conducted based on UniFrac metric (Zhang, Shao & Ye , 2011).

Ethical Statement This article does not contain any studies with human participants or animals performed by any of the authors.

Results

Physicochemical characteristics of the sample sites

Significant differences in physicochemical characteristics among different environmental samples were observed in the present study. The peak NH4+-N content (815.88 mg N kg−1 dry sediment) was detected in WS, which was over 80-folds higher than that in the other three sample sites. And the highest NO2−-N content (14,120 mg N kg−1 dry sediment) was observed in EC, while NO2−-N content in the other three sample site varied form 0.37–127.19 mg N kg−1 dry sediment. Mainly because of its high NO2− content, the highest NOx−-N content was also observed in EC, which was beyond 17-folds higher than that of the other three sample sites. In addition, compared with published researches conducted in paddy fields, where NOx−-N content was around 1.4–3.3 mg N kg−1 dry sediment (Shen et al., 2014; Zhou et al., 2014; Ding et al., 2015) higher NOx−-N content in the agriculture field (PF and CF) of northern China was observed in this study, mainly caused by difference in farming methods.

Abundance of M.oxyfera bacteria

FISH analysis was used to investigate the spatial distribution and relative quantification of M. oxyfera bacteria compared to total bacteria. As shown in Fig. 1, M. oxyfera bacteria (represented by red color) were observed in all four sample sites, and the proportion of M. oxyfera bacteria to total bacteria followed the order of EC > PF > CF > WS. Notably, compared with total bacteria, M.oxyfera bacteria in the enrichment culture took up over 50%, indicating the predominance of M.oxyfera bacteria.

Figure 1 FISH image of the collected samples.

The M. oxyfera bacteria was hybridized with probe S-*-DBACT-1027-a-A-18(Cy3, red) and total bacteria was hybridized with probes EUB I-III (FITC, green). A & E, PF; B & F, CF; C & G, EC, D & H, WS. The scale bar indicates 100 µm.

To further accurately quantify the abundance of M. oxyfera bacteria, qPCR analysis was conducted and significant difference was also observed in different sampling sites. The abundance of M. oxyfera bacteria were 7.28 ± 0.8 × 107, 1.55 ± 0.3 × 107, 1.07 ± 0.3 × 1010, 2.61 ± 0.1 × 106 copies per gram of dry sediment in PF, CF, EC and WS, respectively (Fig. 2). This order was in consistence with results of FISH analysis.

Figure 2 Q-PCR Image of M. oxyfera bacteria.

The abundance of M. oxyfera bacteria 16S rRNA gene copy numbers of collected samples.

Potential rates of n-damo activity

In order to estimate the activity of M. oxyfera bacteria, batch experiments were conducted using the collected samples, and the results are shown in Fig. 3. In experimental groups amended with CH4 and NO2, dramatic decline in CH4 concentration were observed compared with the blank groups, indicating that CH4 oxidation was propelled by NO2− reduction under anoxic conditions. The detected anaerobic methane oxidizing rates were 3.90 ± 0.05, 2.58 ± 0.08, 22.31 ± 0.02 and 1.61 ± 0.01 µmol CH4 g−1 d−1 in PF, CF, EC and WS, respectively. The stoichiometric ratio for methane to nitrite, calculated through the curve fitting method, were 3:5.7 for PF, 3:4.6 for CF, 3:6.9 for EC, and 3:3.2 for WS. The value of n-damo enrichment culture was the closest to the theoretical stoichiometric ratio, which was 3:8 (Ettwig et al., 2010).

Figure 3 Image of batch test.

The consumption rates of methane and nitrite in the paddy field (A), corn field (B), n-damo enrichment culture (C), WWTP (D).

Microbial community structure analysis

In order to determine the microbial community structure of different samples, 454 high-throughput sequencing analysis of 16S rRNA gene was conducted. Raw reads obtained from different samples ranged from 11,017 to 14,814 and the good coverage values varied from 86.48% to 94.70% (Table S2), indicating that these sequences were enough to analyze the microbial community structures. Chao1 estimator, ACE estimator, Shannon index and the numbers of OTUs in four samples followed the same order, which was PF >CF >EC >WS (Table S2).

To show the diversity of species among different samples, rarefaction curves were drawn in this study (Fig. S1). Results showed that the rarefaction curves of all samples didn’t reach a saturation stage, indicating that these samples had highly diverse microbial communities. PCoA was conducted to investigate the differences in microbial community between different samples, based on unweighted UniFrac distance metrics. Results showed that PF and CF tended to cluster together, while EC and WS were obviously different. The results with maximum variation of 87.59% (PC1) and 8.37% (PC2) were shown in Fig. S2.

The difference in microbial community of four samples at the phylum level is shown in Fig. 4. A total of 16 phyla were identified and NC10 was the main phylum observed in PF, CF and EC, accounting for 74.4%, 92.2% and 65.2% of total microorganism, respectively, while Armatimonadetes (formerly candidate division OP10) was the dominant phylum in WS and NC10 phylum only accounted for 7.1% of total microorganism in WS. Since not all NC10 phylum bacteria were defined as M.oxyfera bacteria (Ettwig et al., 2009; Wang et al., 2012) for better understanding and analysis the diversity of M. oxyfera bacteria in different samples, a heat map was conducted at the genera level and the results is shown in Fig. 5. Candidatus.Methylomirabilis bacteria, which were proved to be able to complete n-damo process (Ettwig et al., 2010), accounted for 1.00%, 1.47%, 1.80% and 0.057% of total microorganism in PF, CF, EC and WS, respectively. All these sequences, which were identified as Candidatus.Methylomirabilis, were grouped into 8 (PF), 17 (CF), 9 (EC) and 3 (WS) OTUs at the 97% identity level.

Figure 4 Composition of microbial community at phylum level in different samples.

Figure 5 Richness heat map of the 25 most abundant genera.

Discussion

In present study, PF, CF, EC and WS in northern China, as previously overlooked sites, were selected to investigate the presence and characteristics of n-damo process. Results showed that EC had the highest potential n-damo activity, as well as the highest abundance of M. oxyfera bacteria. Correlation analysis showed that the potential n-damo rates and the abundance of M. oxyfera bacteria followed the same descending order, i.e., EC >  PF >  CF >  WS, indicating positive correlation between the two indexes. Moreover, the potential n-damo rate (22.31 ± 0.02 µmol CH4 g−1d−1) of EC was higher than that reported in other n-damo enrichment culture (He et al., 2014). This was attributed to the relative higher abundance of M. oxyfera bacteria in the present study. The abundance of M. oxyfera bacteria in the present study was over 20 times higher than that reported by He et al. (2014), which further verified the positive correlation between the potential n-damo rates and the abundance of M. oxyfera bacteria.

WWTP showed lower abundance of M. oxyfera bacteria than the other three sample sites, mainly because of its short biomass retention time (13 days), while biomass retention time of other three sample sites was years or even decades of years (Kampman et al., 2014; Weiland, 2006). With the doubling time of 12 weeks (Ettwig et al., 2009), the growth rate of M. oxyfera bacteria is much lower than heterotrophic bacteria, indicating that M. oxyfera bacteria might be washed out in WWTP and resulted in lower abundance of M. oxyfera bacteria. Another possible reason was that high NH4+ content in WWTP, which was unfavorable for the growth of M. oxyfera bacteria. Winkler et al. (2015) found that anammox bacteria had advantage over M. oxyfera bacteria for nitrite in the presence of excess ammonium. What is more interesting, although WS was used as initial inoculum for EC in this study, the abundance of M. oxyfera bacteria in EC was over 4 × 103 times higher than that in WS. This was mainly attributed to a combination of low NH4+ content and high NO2 content during the enrichment period of EC. It was reported that the nitrite affinity constant of M .oxyfera bacteria was 0.6 g NO2–Nm−3, indicating that high NO2− content was more beneficial for the growth of M .oxyfera bacteria (Winkler et al., 2015).

The distribution and composition of NC10 phylum was determined by 16S rRNA gene sequencing analysis. NC10 phylum detected from PF, CF and EC were significant higher than that in WS, which was mainly attributed to short biomass retention time of WWTP In addition, it can be seen from the heat map that the abundance of M. oxyfera bacteria in different samples followed the order of EC > PF > CF >  WS, which was consistence with the result of qPCR. What’s more, PF and CF had much higher OTU numbers than WS. Thus, it was believed that PF and CF could be favorable inoculum for the enrichment of M. oxyfera bacteria, due to their higher abundance and diversity of M. oxyfera bacteria. And it’s worth to notice that significant difference existed in microbial community between EC and WS, EC has six more OTUs than WS, although WS was used as initial inoculum for EC. This might be caused by the optimum enrichment culture for M. oxyfera bacteria in EC, i.e., low NH4+ and high NO2−contents. Besides, the increase in diversity of M. oxyfera bacteria would also be attributed to the longer biomass retention time of EC.

In conclusion, the present study further expanded our knowledge on distribution and characteristic of M. oxyfera bacteria in northern China. Comparative analysis found that positive correlation existed between abundance of M. oxyfera bacteria and potential n-damo activity rate. In addition, PF and CF were identified as suitable inocula to enrich M. oxyfera bacteria. Moreover, long biomass retention time, low NH4+ and high NO2 contents would benefit the growth of M. oxyfera bacteria.

Supplemental Information

Supplemental Information 1 Supplemental Figures and Table

Click here for additional data file.

Data S1 Raw data for Fig. 2

Click here for additional data file.

Data S2 Raw data for Fig. 3

Click here for additional data file.

Data S4 Raw data for Fig. 4

Click here for additional data file.

Data S5 Raw data for Fig. 5

Click here for additional data file.

Additional Information and Declarations

Competing Interests

Author Contributions

DNA Deposition

Data Availability

The authors declare there are no competing interests.

Zhen Hu conceived and designed the experiments, analyzed the data, contributed reagents/materials/analysis tools, wrote the paper, reviewed drafts of the paper.

Ru Ma performed the experiments, analyzed the data, contributed reagents/materials/analysis tools, wrote the paper, prepared figures and/or tables, reviewed drafts of the paper.

The following information was supplied regarding the deposition of DNA sequences:

Sequences obtained from these samples divided into 16S rRNA and pmoA of M. oxyfera were submitted to GenBank under accession numbers KX153190– KX153201 and KX153202– KX153210 respectively.

The following information was supplied regarding data availability:

The raw data has been supplied as Supplementary Files.

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
