# Peer review of "Distribution and characteristic of nitrite-dependent anaerobic methane oxidation bacteria by comparative analysis of wastewater treatment plants and agriculture fields in northern China"

_PeerJ, doi:10.7717/peerj.2766_

## Round 0.1 · original submission · Major Revisions

Please address the reviewers comments. In addition, the written English should need to be improved.

Reviewer 1 ·

Basic reporting

-The English needs much improvement; some sentences are hard to follow and most text is missing a lot of articles (the, and, an/a).
-The introduction lacks, besides proper English, a lot of explanation:
Line 17: which emission factors?
Line 25: what was the main path of AOM?
Line 27: I think this statement needs a reference.
Line 29: presently, researchers are enthusiastic? I think this is quite subjective, please stick to facts.
Line 209: 'This also explained...' I do not believe that potential n-damo rates explained why these were higher than reported rates. Please rephrase.
Line 229: please rewrite, I cannot follow this sentence.
Line 233: 'This was attributed...' I believe that this is not a proven fact, please use words as 'this could be attributed...'
Figures in general are OK, although I believe there is a typo in the legend of figure 3 where the circles represent nitrite concentration of which I believe should be methane concentration.
Figure S1 is doubtful; it shows all phyla but some phyla were not phylogenetically placed and called "others"? I wonder if these sequences are not artifacts. If these are artifacts, they should be omitted from the data.

Experimental design

I think the work that has been done could be valuable since the distribution of M oxyfera in different environments is underexplored and this work touches upon it. However, the research question here is not actually related to this but more on how to enrich for M oxyfera and which inoculum to use. I myself believe this is less relevant to the scientific community and I think it would be much more valuable to try to relate environmental factors to the distribution of M oxyfera in different environments.
Line 35: why is it relevant that information on the distribution of M oxyfera specifically in northern China is lacking? Is it not lacking worldwide?
Line 36: Is it not very obvious that the source of inoculum has great influence on the enrichment culture? I think it is more relevant to know why paddy soils are the optimal inoculum. It is stated that intensive studies on inoculum sources are lacking. I believe it is less relevant to perform these studies. To know which parameters determine growth and activity of M oxyfera is more important and when you know this, you can predict which inoculum is best instead of empirically testing all of them. This is also the main weakness of the paper; the conclusion that is drawn is quite shallow. It only states that some inocula are better than others for enrichment: mainly the inocula where the abundance and activity of M oxyfera is higher. I think this is a very obvious statement and also circular; it does not explain the actual reason why they are more active and abundant.

Line 69: it is stated that samples were pre-incubated to recover microbial activity. Did you add any substrate or electron acceptor? If not, I believe that you lose microbial activity instead of recovering it.
Sampling. At line 48, it is stated that cores were collected by mixing locations? I don't follow this, please explain better. How long were the samples stored before physico-chemical analysis?

Validity of the findings

The data is quite robust and statistically sound. I am quite convinced that most of the data is obtained in a valid fashion and that most data is comprehensive and consistent with each other. Some important points still:
Line 74: it is stated that methane concentrations were 'around' 1 mM. With ‘around’ you mean there is a level of uncertainty? Since you base all incubations on decrease of methane (between 1 and 0 mM), I think it is quite important that these measurements are accurate.
Line 93: PCR amplification. I am a bit puzzled here since it is mentioned that a PCR step was performed but it is not mentioned for what. I assume it is done for the community analysis although it is mentioned that the PCR was nested. This will give quite a bias in the results as this means that at least 2 PCR's were performed.
Line 105: I think the authors should elaborate more on the qPCR method that was used, since this method is quite susceptible to errors. Which controls did you use, etc etc.? Add this in materials and methods. You mention an excellent R square of 0.97. Please avoid subjective terms when describing a method or results. How was the efficiency of the reactions? This is equally important to mention. How many dilutions did you do for the standard curve and did your samples fall within these dilutions so you could interpolate your results?
Line 121: when using blast, you get identity, not similarity indices. This is a very important difference so please correct it.
Line 115: About the phylogenetic trees: I understand that sequencing was done using the miseq platform. What was the average length of your reads that you used for constructing the phylogenetic trees? How long were the other sequences of the database that you used to align with yours? I hope the authors are aware that when you make a tree using sequences of 1500 bp and sequences of only 300-400 bp, this gives biased results and cannot be done. One needs to make the alignment and tree using only 1500 bp sequences and then adding the short reads afterwards when the calculations have been done that determine the tree topology.
As stated above, the article results are OK but the hypothesis and conclusion are weak and could be much improved with additional data. More elaboration on the physico-chemical data would help in explaining the results. For instance, the authors say that ammonium is a determining factor for enrichment of M oxyfera. Any other physico-chemical data on the sampling sites could help as well. These need to be included to be able to draw more meaningful conclusions on distribution of M oxyfera in different environments. Iit is stated that M oxyfera diversity might be related to geographical regions. Why was this concluded? I fail to see how the results support this statement since all samples come from the same river. Also, why was it concluded that M oxyfera distribution is infuenced by salinity? Could you elaborate and use convincing data to support this statement?
Line 145: why is the relative level of enrichment between cultures of different studies compared? There are many parameters that determine how much you can enrich and I believe this statement is therefore not very meaningful.
The activity measurements are all done with 'around' 1 mM methane and results are based on methane decrease. How much methane did you actually add?

Additional comments

General comments:
Abstract. You indicate that the amount of OTUs is different between samples and conclude that the one with a higher amount of OTUs is suitable for growth of M oxyfera. I believe this conclusion cannot be made solely on the amount of OTUs. You have to show relative amounts at least (maybe one sampling site had more OTUs in general). In WWTPs, biomass retention is much lower than in the other environments. Thus, it may well be that M oxyfera grew much faster in the WWTP than in the other samples but it gets washed out.

Reviewer 2 ·

Basic reporting

No Comments

Experimental design

No Comments

Validity of the findings

No Comments

Additional comments

The manuscript of Hu and Ma investigated the distribution and characteristic of nitrite-dependent anaerobic methane oxidation process (n-damo) by comparative analysis of wastewater treatment plants and agriculture fields in northern China. N-damo process is emerging biological nitrogen removal pathway, and is considered as a promising technique in reducing greenhouse gases emissions. Currently, researchers are enthusiastic about investigating the distribution of n-damo process in natural environment. No similar research has been conducted in northern China. In addition, the author identified the optimal inoculum for enrichment of n-damo bacteria through comparative analysis and modern molecular biology technologies. Overall, the experimental design and the results are reasonable and credible, the manuscript could be accepted for publish after minor revision.

Specific comments

1. I think the title is not right. Please correct the title and replace the “nitrite-dependent anaerobic methane oxidation” with “nitrite-dependent anaerobic methane oxidation process”.
2. Page 3, lines 14: replace “WWTP” with “WWTPs”.
3. Page 3, lines 19-20: provide reference for determining “agricultural field is believed to be a more important GHGs sources, mainly because the widely usage of fertilizers”.
4. Page 4, lines 35: change “in environment of northern China” to “in northern China”.
5. In this study, paddy field and corn field were chose to represent agriculture in northern China. Please give the reason.
6. Page 5, lines 49-51: how do you collect the sludge samples?
7. Page 9, lines 69: after washing all the samples, please provide concentrations of NOx- -N and COD.
8. Page 11, lines 138-139: provide data for determining “higher NOx- -N content in the agriculture field (PF and CF) of northern China were observed in this study”.
9. Page 12, lines 161: what is “g” stand for in unit “μmol CH4 g-1 d-1”?
10. Page 16, lines 215: replace “Anammox” by “anammox”.
11. Page 17, lines 238-240: the sentence is ambiguous, please rewrite.
12. Page 17, lines 239: change “import” to “important”.

---

## Round 0.2 · Major Revisions

To a certain extent, the reviewer #1 still disagreed to your replies. Pls carefully address his/her technical comments if you are willing to further revise the manuscript.

Reviewer 1 ·

Basic reporting

The manuscript has improved, but the English grammar still needs attention. I will give only one example of such a sentence, but the whole manuscript needs thorough improvement: "N-damo process is performed by “Candidatus Methylomirabilis oxyfera” (M. oxyfera), affiliated with the NC10 phylum (Ettwig et al. 2010)."
For the rest, I still cannot agree with the following issues:
• line 10: "Comparative analysis revealed that long biomass retention time and optimum environment (low NH4+ and high NO2- content) ..." I do not think that anions are considered environments. Moreover, I still do not agree with the conclusion that distribution and diversity of M. oxyfera bacterial might be related to geographical regions. All the M oxyfera that you find are in close genetic relationship with Yellow River basin sediment. I therefore think you cannot draw that conclusion, but I think you can only say that you have a founder effect.
• Line 104: you stated that you incubated slurries to recover microbial activity. I asked in the previous version if you added any electron acceptor or donor and you stated that you didn’t. I therefore still strongly disagree with the statement that you recover activity. You probably lose it, which is the opposite. The argument that others used the same method is not a valid argument for why you do it.
• You measured DNA concentrations with nanodrop. Did you also measure the concentrations with Nanodrop for qPCR standards and samples? It is well known that nanodrop does not give any reliable quantitative data, fluorescence methods such as Qubit do give it.
• Since the phylogenetic trees are made with short 16S fragments, these trees are highly unreliable and for sure this is normally not done. The authors agree in their rebuttal, and therefore the trees could be removed or it could be explicitly mentioned. I leave this up to the editor.
• The FISH pictures are of very poor quality and I can’t see anything in there.

Experimental design

0

Validity of the findings

0

Reviewer 2 ·

Basic reporting

No Comments

Experimental design

No Comments

Validity of the findings

No Comments

Additional comments

The revised manuscript have been significantly improved, and can be accepted.

---

## Round 0.3 · accepted · Accept

I would like to thank you for choosing PeerJ to publish your research work.